# A Cdc42/RhoA regulatory circuit downstream of glycoprotein Ib guides transendothelial platelet biogenesis

Sebastian Dütting[1,2], Frederique Gaits-Iacovoni[3], David Stegner[1], Michael Popp[1,2], Adrien Antkowiak[3], Judith M.M. van Eeuwijk[1,2], Paquita Nurden[1,4], Simon Stritt[1,2], Tobias Heib[1,2], Katja Aurbach[1,2], Oguzhan Angay[2], Deya Cherpokova[1,2], Niels Heinz[5], Ayesha A. Baig[1,2], Maximilian G. Gorelashvili[1,2], Frank Gerner[1,2], Katrin G. Heinze[2], Jerry Ware[6], Georg Krohne[7], Zaverio M. Ruggeri[8], Alan T. Nurden[4], Harald Schulze[1], Ute Modlich[5], Irina Pleines[1,2], Cord Brakebusch[9] & Bernhard Nieswandt[1,2]

Blood platelets are produced by large bone marrow (BM) precursor cells, megakaryocytes (MKs), which extend cytoplasmic protrusions (proplatelets) into BM sinusoids. The molecular cues that control MK polarization towards sinusoids and limit transendothelial crossing to proplatelets remain unknown. Here, we show that the small GTPases Cdc42 and RhoA act as a regulatory circuit downstream of the MK-specific mechanoreceptor GPIb to coordinate polarized transendothelial platelet biogenesis. Functional deficiency of either GPIb or Cdc42 impairs transendothelial proplatelet formation. In the absence of RhoA, increased Cdc42 activity and MK hyperpolarization triggers GPIb-dependent transmigration of entire MKs into BM sinusoids. These findings position Cdc42 (go-signal) and RhoA (stop-signal) at the centre of a molecular checkpoint downstream of GPIb that controls transendothelial platelet biogenesis. Our results may open new avenues for the treatment of platelet production disorders and help to explain the thrombocytopenia in patients with Bernard–Soulier syndrome, a bleeding disorder caused by defects in GPIb-IX-V.

[1] Institute of Experimental Biomedicine, University Hospital and University of Würzburg, Josef-Schneider-Str. 2, 97080 Würzburg, Germany. [2] Rudolf Virchow Center, University of Würzburg, Josef-Schneider-Str. 2, 97080 Würzburg, Germany. [3] INSERM UMR1048, Institut des Maladies Métaboliques et Cardiovasculaires—I2MC, UMR1048, Institut National de la Santé et de la Recherche Médicale, Université de Toulouse, 1 Avenue Jean Poulhès, BP 84225, 31432 Toulouse Cedex 4, France. [4] Institut Hospitalo-Universitaire LIRYC, Plateforme Technologique d'Innovation Biomédicale, Hôpital Xavier Arnozan, Avenue du Haut Lévêque, 33604 Pessac, France. [5] Research Group for Gene Modification in Stem Cells, LOEWE Center for Cell and Gene Therapy Frankfurt/Main and the Paul-Ehrlich-Institute, Paul-Ehrlich-Straße 51-59, 63225 Langen, Germany. [6] Department of Physiology and Biophysics, University of Arkansas for Medical Sciences, 4301 West Markham Street, Little Rock, Arkansass 72205, USA. [7] Biocenter, University of Würzburg, Am Hubland, 97074 Würzburg, Germany. [8] Department of Molecular Medicine, The Scripps Research Institute, 10550 N Torrey Pines Rd, La Jolla, California 92037, USA. [9] BRIC, Biomedical Institute, University of Copenhagen, Nørregade 10, 1165 Copenhagen, Denmark. Correspondence and requests for materials should be addressed to B.N. (email: bernhard.nieswandt@virchow.uni-wuerzburg.de).

Blood platelets are anucleated cell fragments derived from bone marrow (BM) megakaryocytes (MKs) that are essential for blood clotting. MKs develop from haematopoietic stem cells by a complex maturation process, which includes DNA replication without cell division (endomitosis) and the formation of a demarcation membrane system (DMS), which functions as a membrane reservoir for newly formed platelets[1,2]. Mature BM MKs are giant, polyploid cells that localize close to BM sinusoids in order to extend and release long cytoplasmic protrusions called proplatelets into the sinusoidal lumen, from which platelets are shed under the influence of blood shear forces[3–6]. The highly controlled process of transendothelial platelet biogenesis stands in contrast to transendothelial migration of whole haematopoietic (progenitor) cells during mobilization and homing[7]. A current concept incorporates the idea that the chemokine stromal cell-derived factor-1 triggers migration of early MKs from the endosteal niche towards BM sinusoids[8,9] and that the lipid mediator sphingosine-1-phosphate elicits directed proplatelet extension into the circulation[10]. However, the molecular mechanisms regulating the trafficking of MKs and haematopoietic cells within the BM and across the endothelial barrier are poorly defined.

Rho GTPases are small proteins (20–25 kDa) belonging to the superfamily of Ras-related proteins which are found in all eukaryotic cells[11]. They are best known for regulating cytoskeletal dynamics in virtually all cell types[12]. The best-characterized Rho GTPases are Cdc42, RhoA and Rac1, whose activation is associated with the formation of filopodia, stress fibres and lamellipodia, respectively. We previously reported that transgenic mice lacking either Cdc42 or RhoA in MKs and platelets exhibit pronounced macrothrombocytopenia, indicating a distinct role of these molecules in platelet production[13,14]. In addition, Cdc42 deficiency led to decreased filopodia formation of platelets on von Willebrand factor (vWF), suggesting a unique role of Cdc42 downstream of the glycoprotein (GP) Ib subunit of the vWF receptor complex GPIb-IX-V (ref. 13). Bernard–Soulier syndrome (BSS) is a rare platelet disorder characterized by macrothrombocytopenia, which is caused by damaging variants in either of the three genes encoding the GPIbα/β or GPIX subunits of the GPIb-IX-V complex, leading to its absence from, or dysfunction at the MK and platelet surface[15–19]. The mechanisms by which the receptor controls megakaryopoiesis and platelet production are largely unknown.

Here, we show that lack of either functional GPIb or Cdc42 reduced MK polarization in vitro and impaired MK localization at sinusoids and transendothelial biogenesis in vivo. In contrast, absence of RhoA in MKs resulted in increased Cdc42 activity and GPIb-dependent transendothelial migration of entire MKs into BM sinusoids. These results reveal that Cdc42 and RhoA act as a regulatory circuit downstream of GPIbα to coordinate MK polarization and transendothelial platelet biogenesis in vivo.

## Results

**GPIbα signalling controls MK localization.** Lack of functional GPIb-IX-V results in impaired DMS development and distribution during MK maturation in humans and mice[20,21]. We analysed MK localization in the BM of mice lacking GPIbα ($Gp1ba^{-/-}$) by immunostaining of native cryosections of whole femora (Fig. 1). In wild-type (wt) mice, the majority of MKs was in close contact to BM sinusoids ($61.5 \pm 4.7\%$; Fig. 1a,b). Only rarely were MKs found within the sinusoidal lumen ($0.6 \pm 1.4\%$). Interestingly, in $Gp1ba^{-/-}$ mice the number of MKs with direct contact to sinusoids was significantly reduced ($48.1 \pm 4.0\%$; $P = 0.0002$; two-way ANOVA with Bonferroni correction for multiple comparisons) resulting in an increase in BM

haematopoietic compartment (BMHC)-localized MKs ($50.2 \pm 6.3\%$; $P = 0.0005$; Fig. 1a,b).

In view of the multiple intrinsic defects present in MKs deficient in GPIbα[22] or GPIbβ[23] we turned to a mouse line expressing a mutant version of GPIbα, where the ectodomain of GPIbα is replaced by that of the human interleukin-4 receptor α (IL-4Rα) ($Gp1ba^{-/-;tg}$, further referred to as $Gp1ba$-$Tg$). In these mice, the BSS-associated macrothrombocytopenia is ameliorated, but not fully reversed (Supplementary Fig. 1a,b (refs 21,24)), implying a specific role for the ectodomain of GPIbα in platelet biogenesis. Strikingly, the BM MK distribution in $Gp1ba$-$Tg$ mice and $Gp1ba^{-/-}$ mice was similar, with a significant decrease in the proportion of MKs with sinusoidal contact (wt: $67.5 \pm 6.4\%$; $Gp1ba$-$Tg$: $49.4\% \pm 7.0\%$; $P < 0.0001$) and a concomitant increase of BMHC-localized MKs (wt: $30.3 \pm 6.6\%$; $Gp1ba$-$Tg$: $49.0 \pm 6.0\%$; $P < 0.0001$; two-way ANOVA with Bonferroni correction for multiple comparisons) (Fig. 2a,c). The role of the GPIbα ectodomain for MK localization was further confirmed by treatment of wt mice with the monovalent Fab fragment of an antibody directed against the major ligand (vWF)-binding domain of GPIbα (p0p/B-Fab) that is known not to affect platelet survival in the circulation[25]. While GPIbα blockade had no effect on total MK numbers in the BM (Supplementary Fig. 1c), it clearly reduced the fraction of MKs with direct sinusoidal contact ($39.5 \pm 4.4\%$; $P < 0.0001$; two-way ANOVA with Bonferroni correction for multiple comparisons, Fig. 2a,c). This was associated with a reduction in peripheral platelet counts and an increase in platelet size by about one third (Fig. 2d,e) as compared to wt, similar to that seen in $Gp1ba$-$Tg$ mice (Supplementary Fig. 1a,b). Formation of the DMS, as analysed by transmission electron microscopy (TEM), was unaltered in MKs of $Gp1ba$-$Tg$ mice or wt mice after GPIbα blockade (Fig. 2b). Together, these findings revealed a critical role for the ectodomain of GPIbα in controlling MK localization at vascular sinusoids, independent of its role in DMS development and partitioning.

We next sought to get insights into the signalling that triggers GPIbα-dependent MK guidance within the BM. Using mice lacking Cdc42 in MKs and platelets ($Cdc42^{fl/fl\ Pf4-cre}$, further referred to as $Cdc42^{-/-}$ (ref. 13)), we have previously shown that Cdc42 controls cytoskeletal dynamics downstream of GPIbα in platelets and that its absence in MKs causes marked macrothrombocytopenia by unknown mechanisms[13]. Intriguingly, significantly fewer $Cdc42^{-/-}$ MKs were in direct contact with BM sinusoids in line with an increased MK population in the BMHC (Fig. 2f,g) compared to wt littermate controls ($52.0 \pm 3.6$ $P = 0.0002$; two-way ANOVA with Bonferroni correction for multiple comparisons). Importantly, $Cdc42^{-/-}$ MKs displayed only partially reduced membrane invaginations, indicating that the thrombocytopenia in these mice was not solely the result of a lack of intracellular membranes (Fig. 2h). Another key component of the GPIbα signalling machinery is phosphoinositide 3-kinase (PI3K) that has been shown to play an important role in thrombosis and GPIbα-induced integrin activation[26,27]. Of note, treatment of wt mice with the PI3K inhibitor wortmannin reduced the number of MKs in direct contact with sinusoids compared to the control ($55.0 \pm 5.9\%$, $P = 0.003$; two-way ANOVA with Bonferroni correction for multiple comparisons) (Supplementary Fig. 2d,e), whereas it had no effect on DMS development or total MK numbers in the BM (Supplementary Fig. 2a,b,f).

In neuronal cells, PI3K-mediated establishment of cell polarity was shown to involve the interplay of GTPases with atypical protein kinase C (aPKC) isoforms, which form part of the PAR ('partitioning defective') complex[28]. We observed reduced sinusoidal localization of MKs in mice lacking the aPKC isoform

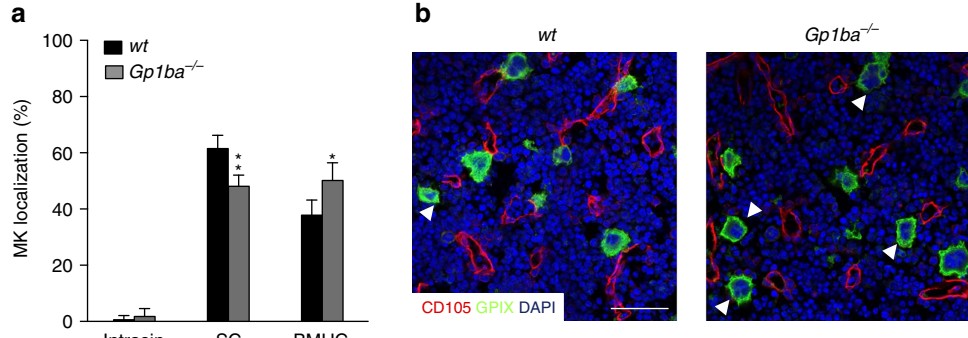

**Figure 1 | GPIbα deficiency alters MK localization in the BM.** (**a**) Quantification of MK localization in the BM reveals less sinusoidal contact (SC) and increased localization in the BM haematopoietic compartment (BMHC) in Gp1ba$^{-/-}$ mice (grey) compared to the wild-type (wt, black) (n = 5); intrasinusoidal (intrasin.). (**b**) Representative confocal images of immunostained BM of wt and Gp1ba$^{-/-}$ mice. Scale bars, 50 µm. MKs, proplatelets and platelets are shown by CD41 staining in green. Endoglin staining (red) labels vessels. DAPI, blue. Arrowhead indicates MKs in the BM haematopoietic compartment (BMHC). Bar graphs represent mean ± s.d. Two-way ANOVA with Bonferroni correction for multiple comparisons; **P < 0.01; *P < 0.05.

PKCι in MKs/platelets (Supplementary Fig. 3a,b), indicating a role of the PAR complex and atypical PKCs for PI3K-Cdc42 controlled MK polarization and migration/localization. Regulation of cell polarization by the PAR complex is characterized by a profound reorganization of the microtubule cytoskeleton[29]. Consistently, we found that microtubule disruption by nocodazole treatment prevented transmigration of RhoA-deficient MKs (Supplementary Fig. 3c,d, see below) similar to Cdc42-deficiency.

**RhoA negatively regulates GPIb-Cdc42 signalling in MKs.** Rho GTPases integrate phosphoinositide signalling and cytoskeletal dynamics. This involves a cross-regulation between Cdc42 and RhoA, which tunes GTPase function in different cell types[30,31]. It was recently demonstrated that Cdc42 is important to polarize the MK DMS towards BM sinusoids[32]. Moreover, we have previously shown that mice with MK/platelet-specific RhoA-deficiency (RhoA$^{fl/fl}$ $^{Pf4-cre}$, further referred to as RhoA$^{-/-}$) exhibit pronounced macrothrombocytopenia (Supplementary Fig. 4a,b). This finding was striking since it was reported that RhoA negatively regulates proplatelet formation of human CD34 + derived MKs in vitro[33]. Supporting this result, we found that proplatelet formation of RhoA$^{-/-}$ fetal liver cell-derived MKs in liquid culture in vitro was comparable to the wt (Supplementary Fig. 4e (ref. 14)). Taking these observations together, we hypothesized that the apparent platelet biogenesis defect in RhoA$^{-/-}$ mice might be mechanistically linked to defective control of GPIbα-Cdc42 signalling and MK localization within the BM. Intriguingly, in BM sections of RhoA$^{-/-}$ mice, we found a dramatic MK mislocalization with approximately 30% of the cells being present inside the BM sinusoids (27.9 ± 10%, P < 0.0001; two-way ANOVA with Bonferroni correction for multiple comparisons; Fig. 3a,b, Supplementary Fig. 4c,d). The remaining MKs were mostly found in direct contact with sinusoids (61.2 ± 8.6%, P = 0.430; two-way ANOVA with Bonferroni correction for multiple comparisons), while only a very minor population was present in the BMHC (10.8 ± 2.3%, P < 0.0001; two-way ANOVA with Bonferroni correction for multiple comparisons). To analyse the ultrastructure of intrasinusoidal MKs in the intact BM, we performed TEM on cross-sections of intact femora (Fig. 3c, Supplementary Fig. 5). Intrasinusoidal RhoA$^{-/-}$ MKs showed signs of maturation including poly-lobulated nuclei. The MKs either contained a large quantity of cytoplasm with a well-developed membrane system and a pronounced peripheral zone, or showed signs of flow-dependent DMS/cytoplasm detachment suggestive of proplatelet

formation. Nevertheless, most MKs remained in contact with sinusoidal endothelial cells. Of note, DMS formation appeared mostly unaffected in RhoA$^{-/-}$ MKs when analysed by TEM on BM cross-sections (Supplementary Fig. 5c), but we frequently found intact cells of other lineages (typically granulocytes) within the MKs cytoplasm (Fig. 3c, Supplementary Fig. 5d), a phenomenon known as emperipolesis[34].

Next, we utilized intravital two-photon microscopy (2PM) for dynamic visualization of MK localization and proplatelet formation in the BM of the skull over time. As previously described[4], in wt mice multiple MKs were in direct sinusoidal contact and eventually released long proplatelets into the vascular sinus (Fig. 3d–f, Supplementary Movie 1). In sharp contrast, in RhoA$^{-/-}$ mice, intravital proplatelet formation was increased (Supplementary Fig. 6a) and entire MKs (i) transmigrated from the BM compartment into the vascular sinus (Fig. 3e, Supplementary Movie 2, Supplementary Fig. 6b), (ii) adhered for up to 30 min (maximal observation period) inside the vascular lumen (Fig. 3f, Supplementary Movies 3 and 4) or (iii) extended and eventually released very large proplatelet-like fragments into the circulation (Supplementary Fig. 6c, Supplementary Movie 5). Although the pulmonary microvessels represent the first capillary bed that receives circulating blood from the BM[35], we did not observe increased numbers of MKs in the lungs of RhoA$^{-/-}$ mice (Supplementary Fig. 6d,e), suggesting that the intrasinusoidal MKs predominantly fragment in the BM vasculature or elsewhere in the circulation[3]. These observations collectively indicate that RhoA negatively regulates MK guidance/localization towards BM vascular sinusoids and thereby acts as a key regulator of proplatelet extension across the endothelial barrier by providing a stop-signal for MK transmigration.

To assess whether the transmigration of RhoA$^{-/-}$ MKs was indeed GPIbα-dependent, we blocked the major ligand-binding site of the receptor in wt and RhoA$^{-/-}$ mice by injection of p0p/B-Fab. Strikingly, whereas this treatment did not affect overall MK numbers in the BM (Supplementary Fig. 7), it reduced the intrasinusoidal localization of RhoA$^{-/-}$ MKs by more than 70% (8.2 ± 4.4%, P < 0.001; two-way ANOVA with Bonferroni correction for multiple comparisons, Fig. 4a,d). This was associated with an increased presence of MKs in direct contact with BM sinusoids and a decreased percentage of BMHC-localized MKs compared with wt controls and vehicle-treated RhoA$^{-/-}$ mice, respectively (Fig. 4d). In contrast, treatment with Fab fragments of antibodies directed against the major platelet integrin αIIbβ3 (JON/A (ref. 36)) or GPV (DOM1 (ref. 37)) did neither affect MK localization in the wt

(Supplementary Fig. 7b,c), nor was it able to revert the transendothelial migration of $RhoA^{-/-}$ MKs, demonstrating a specific role for GPIb in these processes. We also generated

$RhoA^{-/-}/Gp1ba$-$Tg$ mice, in which we found similar results as compared to p0p/B-Fab treatment (9.5 ± 4.5% MKs intra-sinusoidal, Fig. 4b,d). Interestingly, inhibition of PI3K by

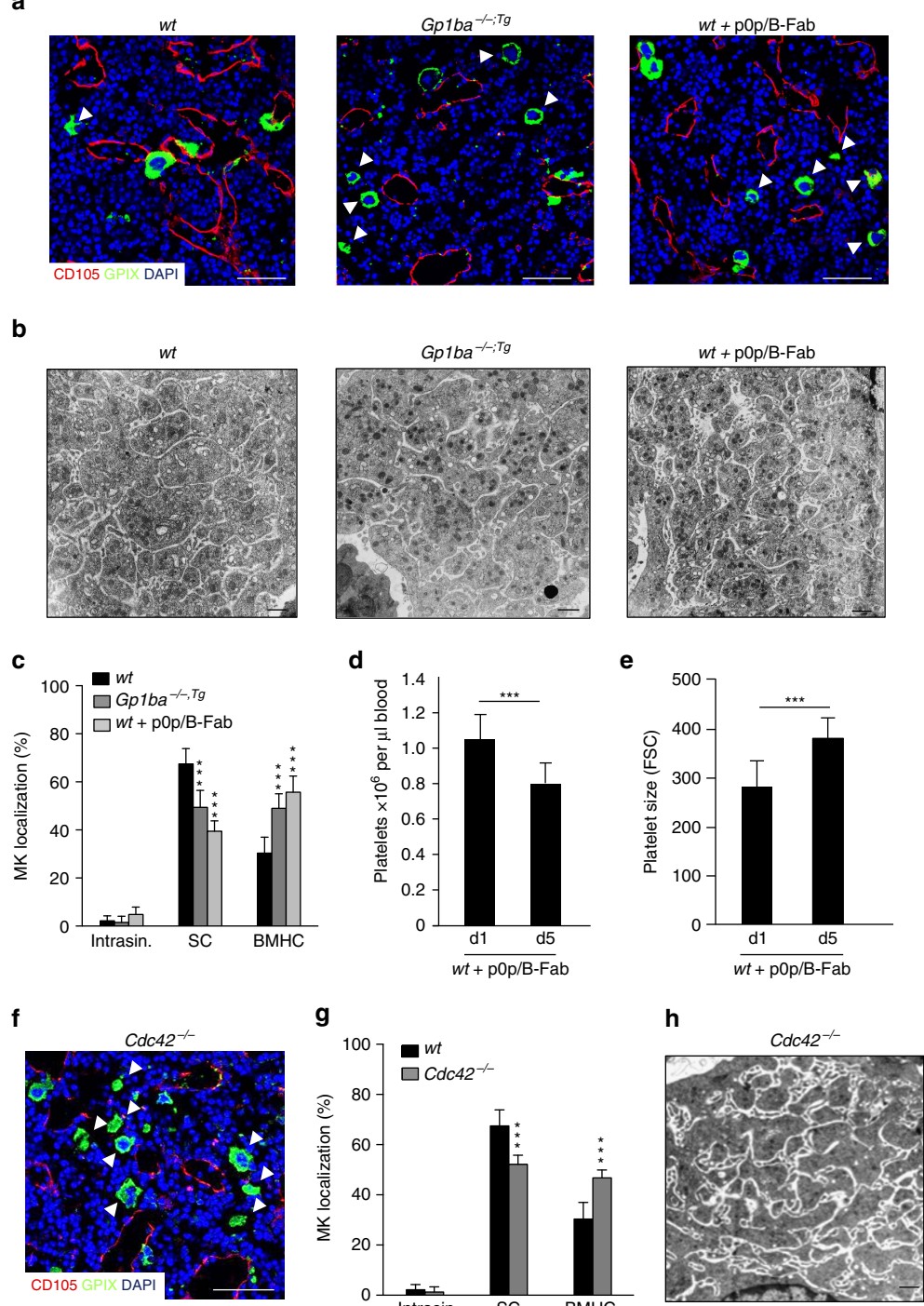

**Figure 2 | GPIbα ectodomain and Cdc42 regulate MK localization in the BM.** (**a**) Confocal images of immunostained BM and (**b**) TEM analysis of BM MKs of *wt* (*left panel*), $Gp1ba^{-/-;tg}$ (*Gp1ba-Tg*) (*middle panel*) and *wt* mice after treatment with GPIbα-blocking monovalent Fab fragments, p0p/B-Fab (*right panel*). $n=4$ biological replicates. Scale bars, 50 μm (**a**) and 2 μm (**b**). MKs, proplatelets and platelets are shown by GPIX staining in green. Endoglin staining (red) labels vessels. DAPI, blue. Arrowhead indicates MKs in the BM haematopoietic compartment (BMHC). (**c**) Quantification of MK localization in the BM reveals less sinusoidal contact (SC) in *Gp1ba-Tg* (dark grey) and *wt* mice after GPIbα-blockade (light grey) compared to the *wt* (black) ($n=7$, 5 and 10); intrasinusoidal (intrasin.). (**d,e**) Reduced platelet count (**d**) and increased platelet size (**e**) in *wt* mice after GPIbα-blockade ($n=4$). (**f**) Representative confocal images of immunostained BM of $Cdc42^{-/-}$ mice. Scale bar, 50 μm. (**g**) Quantification of MK localization in the BM reveals reduced SC in *wt* (black) and $Cdc42^{-/-}$ (grey) mice ($n=10$ and 3). (**h**) TEM analysis of BM MKs of $Cdc42^{-/-}$ mice. Scale bar, 2 μm. Bar graphs represent mean ± s.d. (**c,g**) Two-way ANOVA with Bonferroni correction for multiple comparisons; (**d,e**) Unpaired two-tailed Student's *t*-test; ***$P<0.001$.

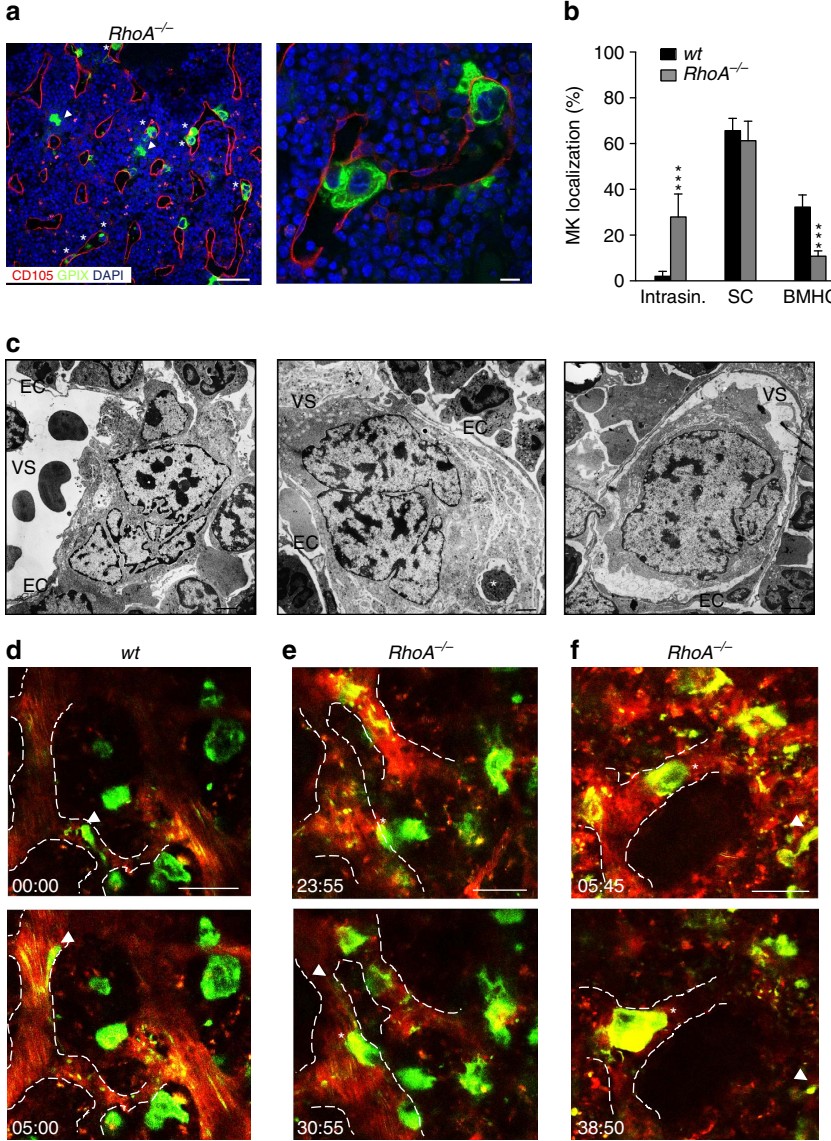

**Figure 3 | RhoA negatively regulates MK guidance and transmigration.** (**a**) Representative confocal images of immunostained BM of *RhoA*<sup>−/−</sup> mice. Scale bars, 50 μm (left panel); 10 μm (right panel). MKs, proplatelets and platelets are shown by GPIX staining in green. Endoglin staining (red) labels vessels. DAPI, blue. Arrowhead indicates MKs in BMHC, asterisk indicates intrasinusoidal (intrasin.) MKs. (**b**) Quantification of MK localization in the BM reveals abundant intrasinusoidal localization of *RhoA*<sup>−/−</sup> (grey) MKs compared to the *wt* (black) (*n* = 9). (**c**) TEM analysis of *RhoA*<sup>−/−</sup> BM MKs. Scale bars, 2.5 μm. EC, endothelial cell; VS, vascular sinus. Asterisk indicates emperipolesis (*n* = 6 (wt) and 8 (*RhoA*<sup>−/−</sup>)). (**d–f**) Intravital two-photon microscopy of *wt* (**d**) and *RhoA*<sup>−/−</sup> (**e**,**f**) BM MKs in the skull. Arrowhead indicates proplatelet formation, asterisk indicates intrasinusoidal MKs (*n* = 7 and 6). Scale bars, 50 μm. Bar graphs represent mean ± s.d. Two-way ANOVA with Bonferroni correction for multiple comparisons; ***P < 0.001.

wortmannin treatment likewise significantly reduced the number of intrasinusoidal MKs in *RhoA*<sup>−/−</sup> mice (12.3 ± 5.2%, *P* < 0.0001; two-way ANOVA with Bonferroni correction for multiple comparisons), but did not increase the number of MKs with direct sinusoidal contact (Supplementary Fig. 2d,e). Total MK numbers in the BM were unaltered in wortmannin-treated compared to vehicle-treated *RhoA*<sup>−/−</sup> mice (Supplementary Fig. 2f) and also their DMS appeared largely unaltered (Supplementary Fig. 2c).

Together, these data indicate that RhoA may act as a negative regulator of GPIbα-Cdc42-driven MK guidance towards BM sinusoids and transendothelial migration. To further test this hypothesis, we generated MK/platelet-specific RhoA/Cdc42-double-deficient mice (*RhoA/Cdc42*<sup>fl/fl Pf4-cre</sup>, further referred to as *RhoA/Cdc42*<sup>−/−</sup>; Supplementary Figs 8a and 11a–c). Strikingly, *RhoA/Cdc42*<sup>−/−</sup> MKs massively accumulated around

vascular sinusoids (82.9 ± 5.9%, *P* = 0.0002; two-way ANOVA with Bonferroni correction for multiple comparisons, Fig. 4c–e, Supplementary Movie 6), but failed to transmigrate into the lumen (3.9 ± 1.8% MKs intrasinusoidal, Fig. 4c,e) or to efficiently form transendothelial proplatelets *in vivo* (Fig. 4f,g, Supplementary Movie 6). Consequently, these animals were severely thrombocytopenic (Supplementary Fig. 8b,c). Of note, intrasinusoidal localization of *RhoA*<sup>−/−</sup> MKs was not altered by the concomitant lack of Rac1 (*RhoA/Rac1*<sup>−/−</sup>) (Supplementary Fig. 9a–e), thus indicating a specific RhoA-Cdc42 crosstalk downstream of GPIbα in this process.

**RhoA controls MK polarization by limiting Cdc42 activity.** We next investigated whether a possible crosstalk between Cdc42 and RhoA regulates MK polarization. Importantly, mature

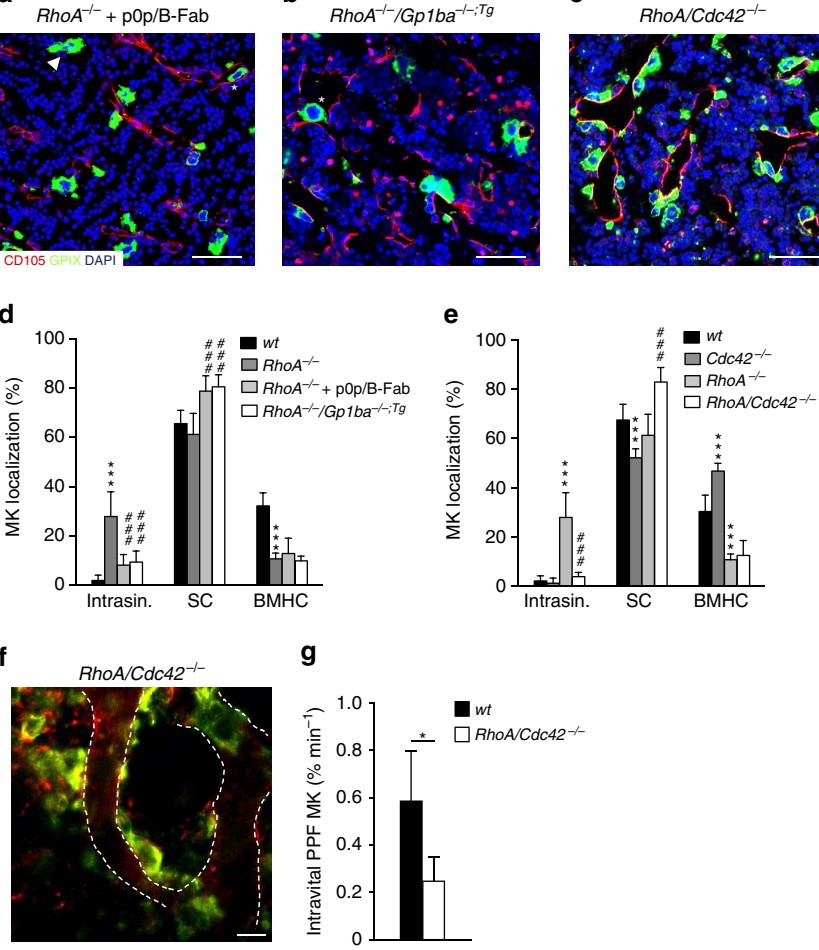

**Figure 4 | RhoA restricts GPIbα-Cdc42-driven MK localization and transendothelial migration.** (**a–c**) Representative confocal images of immunostained BM of $RhoA^{-/-}$ mice after treatment with GPIbα-blocking monovalent Fab fragments, p0p/B-Fab (**a**), concomitant lack of the GPIbα ectodomain (**b**), or $RhoA/Cdc42^{-/-}$ mice (**c**). $n = 4$ biological replicates. Scale bars, 50 μm. MKs, proplatelets and platelets are shown by GPIX staining in green colour. Endoglin staining (red) labels vessels. DAPI, blue. Arrowhead indicates MKs in BMHC, asterisk indicates intrasinusoidal (intrasin.) MKs. (**d**) Quantification of MK localization in the BM of $RhoA^{-/-}$ mice reveals reduced intrasinusoidal localization upon concomitant blockade (light grey) or absence (white) of GPIbα compared to normal $RhoA^{-/-}$ mice (dark grey) ($n = 9$, 3 and 9). Untreated wt is shown in black ($n = 9$ biological replicates). (**e**) Quantification of MK localization in the BM of $RhoA/Cdc42^{-/-}$ mice reveals reduced intrasinusoidal localization compared to $RhoA^{-/-}$ mice (light grey) and MK clustering at sinusoids. Wt (black): $n = 10$, $Cdc42^{-/-}$ mice (dark grey); $n = 3$. (**f**) Intravital two-photon microscopy of wt (black) and $RhoA/Cdc42^{-/-}$ MKs (white) in the skull ($n = 7$ and 4). Scale bar, 50 μm. (**g**) Quantification reveals reduced proplatelet-formation (ppf). Bar graphs represent mean ± s.d. (**d,e**) Two-way ANOVA with Bonferroni correction for multiple comparisons; (**g**) Unpaired two-tailed Student's $t$-test; *$P < 0.05$; ***$P < 0.001$ compared to wt; ###$P < 0.001$ compared to $RhoA^{-/-}$.

$Cdc42^{-/-}$ and $Gp1ba$-Tg BM MKs displayed defective DMS polarization after 4 days of liquid culture *in vitro* (Fig. 5a–c,f), whereas increased polarization of not fully mature DMS was observed in $RhoA^{-/-}$ MKs (Fig. 5d,g). Strikingly, *in vitro* treatment with p0p/B-Fab (anti-GPIb) markedly reduced DMS polarization in cultures of both wt and $RhoA^{-/-}$ BM MKs (Fig. 5e,g), suggesting that the regulatory role of GPIbα in this process depends on intracytoplasmic GPIb-IX-V signalling and may operate in the absence of an ectopic ligand.

To decipher the underlying mechanism, we took advantage of lentiviral FRET biosensors derived from the Raichu probe to monitor Cdc42 activity in the polarized DMS[32]. Wt MKs showed a polarized Cdc42 activity at the DMS/F-actin complex, which was reduced by 68% in $Gp1ba$-Tg MKs (Fig. 6a,b,d), demonstrating the importance of controlled Cdc42 activation by GPIbα for correct DMS polarization. Strikingly, polarized Cdc42 activity showed a 2.4-fold increase in the absence of RhoA

(Fig. 6c,d), indicating that RhoA controls DMS and subsequently MK polarization by limiting Cdc42 activity. In order to determine how the nucleotide state of Cdc42 and RhoA influences this regulation, we utilized single amino acid mutants Cdc42/F28L and RhoA/F30L known to induce a constitutive exchange of GDP for GTP, thus resulting in hyperactive GTPases[38]. To restrict the expression of these mutants to the megakaryocytic lineage, we generated BM chimeric mice by transplantation of HSCs after lentiviral gene transfer, in which proteins were expressed under transcriptional control of the MK/platelet-specific human *GP6* promoter (Fig. 7a,b)[39]. Of note, control mice transplanted with GFP-expressing HSCs showed reduced sinusoidal MK localization compared with non-irradiated mice (44.3 ± 2.94%, Fig. 7c,f), indicating that irradiation influences MK localization. Compared to GFP-positive control cells, vessel association of Cdc42/F28L expressing MKs (GFP+/GPIX+) was increased (57.0 ± 9.0%, Fig. 7c,d,f) and approx. 4% of mutant MKs were

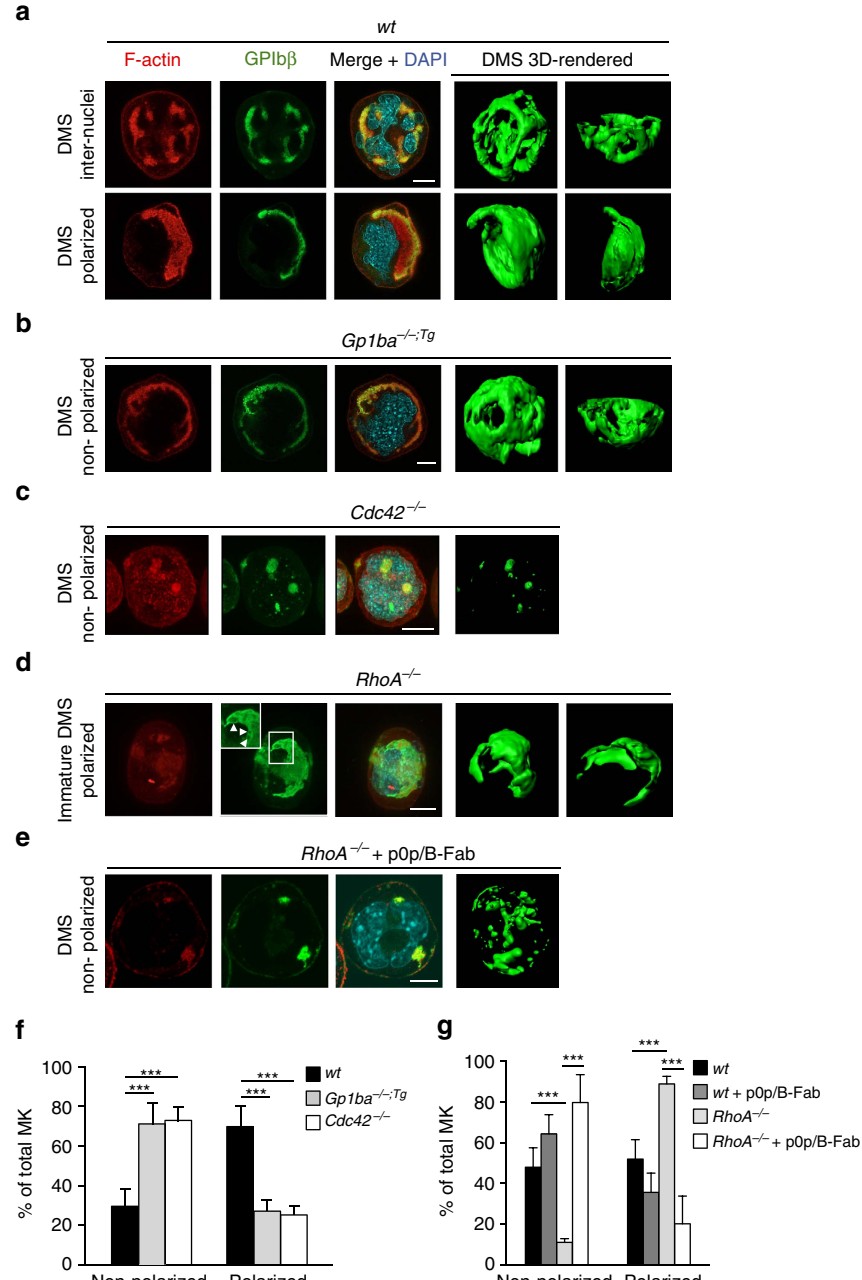

**Figure 5 | RhoA controls GPIbα/Cdc42-dependent MK polarization.** (**a**–**e**) Representative confocal images of *in vitro* differentiated MKs reveal polarized DMS in mature *wt* MKs (**a**), defective polarization in $Gp1ba^{-/-;Tg}$ ($Gp1ba$-$Tg$) (**b**) and $Cdc42^{-/-}$ MKs (**c**), increased polarization in $RhoA^{-/-}$ MKs (**d**) and defective polarization of $RhoA^{-/-}$ MKs after treatment with GPIbα-blocking monovalent Fab fragments, p0p/B-Fab (**e**). F-actin is stained by phalloidin (red); GPIbβ (**a**–**d**) or GPIX (**e**), green; DAPI, blue. 3D surface rendering of respective z-stacks (Imaris software; right panel). $n = 5$ biological replicates. Scale bar: 10 μm. (**f**) Quantification of DMS polarization shows reduced MK polarization in $Gp1ba$-$Tg$ (light grey) and $Cdc42^{-/-}$ (white) MKs compared to the *wt* (black) ($n = 5$ biological replicates). (**g**) MK polarization is increased in $RhoA^{-/-}$ MKs (light grey) compared to the untreated *wt* (black) and wt MKs treated with p0p/B-Fab fragments (dark grey). Treatment of $RhoA^{-/-}$ MKs with p0p/B-Fab fragments (white) reverts the hyperpolarization. $n = 5$ biological replicates (*wt* and $RhoA^{-/-}$) and 2 biological replicates (*wt* + p0p/B-Fab and $RhoA^{-/-}$ + p0p/B-Fab). Bar graphs represent mean ± s.d. Two-way ANOVA with Bonferroni correction for multiple comparisons; ***$P < 0.001$.

located inside the BM sinusoids, whereas none were found in this compartment in control mice (Fig. 7c,d,f). In contrast, RhoA/F30L expression resulted in a further reduction of MKs in direct contact with BM sinusoids ($30.4 \pm 6.4\%$, Fig. 7c,e,f) confirming the opposing functions of active Cdc42 (go-signal) and RhoA (stop-signal) in controlling MK localization and transendothelial proplatelet formation (Fig. 8).

## Discussion

MKs are unique among haematopoietic progenitor cells in that they normally do not cross the endothelial lining of BM sinusoids, but extend and release portions of their cytoplasm into the blood stream by so far unknown mechanisms[4,5]. We now provide compelling evidence that this key step of unidirectional transendothelial proplatelet formation is tightly controlled

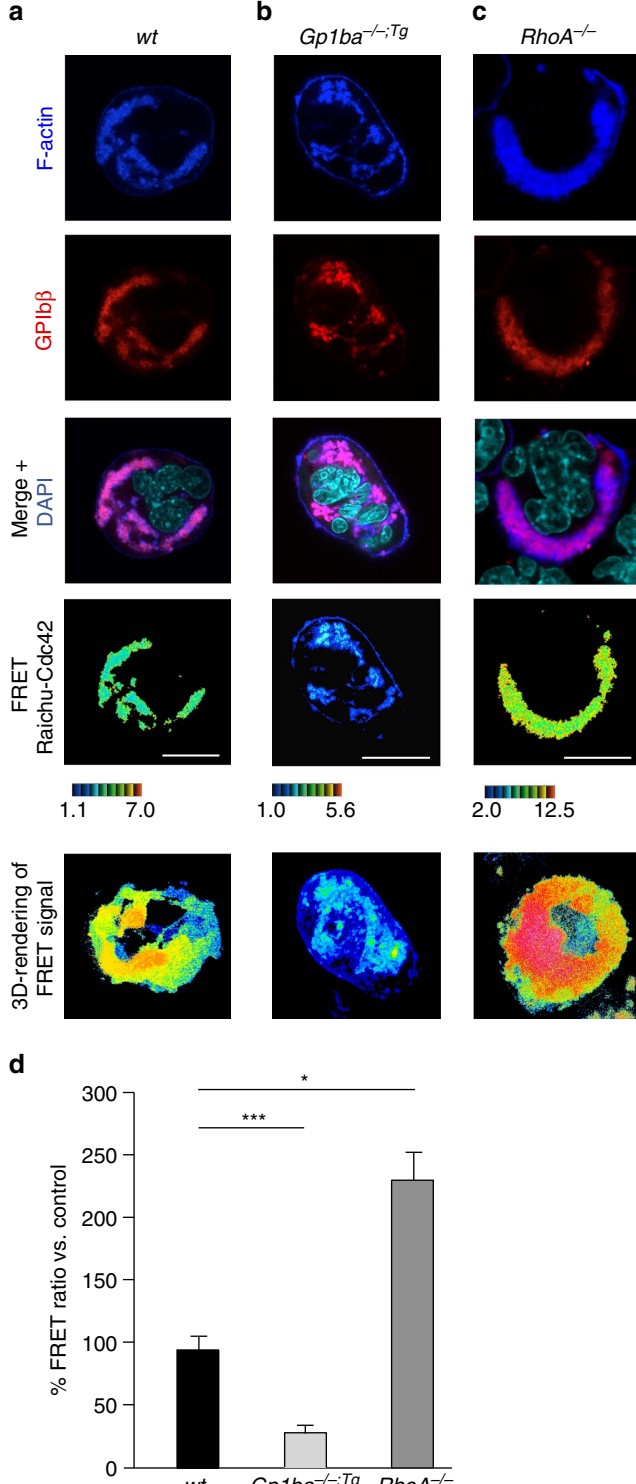

**Figure 6 | RhoA controls GPIbα-induced MK polarization by limiting Cdc42 activity.** (**a–c**) Cultured mouse MKs were transduced with Raichu-Cdc42 lentiviral vector. F-actin was stained with phalloidin (blue); GPIbβ, red; DAPI, cyan. Cdc42 activation was visualized by colour-coded FRET. High ratio values (red) correlate with higher Cdc42 activity. Active Cdc42 localizes with the polarized F-actin/DMS complex and its activity is decreased in $Gp1ba^{-/-;Tg}$ ($Gp1ba$-$Tg$) and increased in $RhoA^{-/-}$ MKs. Scale bar: 10 μm. (**d**) Quantification reveals decreased Cdc42 activity in $Gp1ba^{-/-;Tg}$ (light grey) and increased activity in $RhoA^{-/-}$ (dark grey) polarized MKs compared to the $wt$ (black) ($n = 4$ biological replicates). Bar graphs represent mean ± s.d. Two-way ANOVA with Bonferroni correction for multiple comparisons; *$P < 0.05$; ***$P < 0.001$.

within MKs by a crosstalk of Cdc42 and RhoA and requires a functional GPIbα ectodomain to occur efficiently (Fig. 8). Our data reveal a central antagonistic role of Cdc42 and RhoA in MK polarization and suggest that absence of RhoA supports locally increased Cdc42 activity and thereby directs the sites of cell protrusion. Our findings also highlight a critical function of the GPIbα ectodomain for efficient MK polarization, Cdc42 activation and thus sinusoidal localization and transendothelial proplatelet formation of mature MKs. Thus, these data indicate that MK mislocalization within the BM contributes to the low platelet counts in BSS patients and may mechanistically explain the defective transendothelial (pro-) platelet formation by impaired DMS polarization and Cdc42 activation.

Of note, vWF-deficient humans and mice display normal platelet counts[40], and MK localization in the BM was not altered in $Vwf^{-/-}$ mice (Supplementary Fig. 10a,b), which excludes a major role of this ligand in GPIbα-dependent MK polarization.

Besides the vWF binding site, the p0p/B antibody Fab fragment also completely blocks thrombin binding to mouse GPIbα (Supplementary Fig. 10c), raising the possibility that this process might play a role in MK localization/polarization. However, MK localization in the BM of mice expressing human GPIbα with a mutation (D277N) that abolishes α-thrombin binding[41] was similar to that of mice expressing $wt$ hGPIbα (Supplementary Fig. 10d,e), arguing against a role of thrombin in this process. Intriguingly, however, $in$ $vitro$ treatment of cultured MKs with p0p/B-Fab was sufficient to almost completely revert the DMS hyperpolarization observed in RhoA-deficient MKs. Thus, our findings do not indicate that GPIbα requires binding of an ectopic ligand to control MK localization and polarization in the BM. Rather, our results support the hypothesis that GPIbα-mediated regulation of MK polarization might be a cell intrinsic process, which can be modulated by altering GPIbα signalling through changing the conformation and/or membrane localization/clustering of the receptor in response to p0p/B-Fab binding. In line with this, enhanced endogenous binding of mutant vWF to GPIbα in patients with vWF disease-type 2B results in giant platelets and thrombocytopenia and was reported to be associated with dysregulation of the LIM kinase/cofilin pathway in MKs, together with upregulated RhoA signalling[42]. Thus, altered GPIb-IX-V-mediated intracellular signalling can lead to abnormal cytoskeletal signalling and severely abnormal MK function.

Taken all observations together, our data support a model in which GPIbα controls MK localization and transendothelial MK migration through intracellular signals, namely through the regulation of cell polarization via the Rho GTPases RhoA and Cdc42. While this process presumably involves MK PI3K function, we cannot rule out that treatment with the PI3K inhibitor wortmannin also exerts effects on vascular cells. Thus, deciphering the exact role of PI3K and/or PIs in MK localization requires further investigation. On the other hand, it is well known that Cdc42 and PI3K play a central role in establishing morphological and molecular cell polarity in eukaryotic cells[28,29]. In neurons, establishment of cell polarity during axon specification requires PI3K-dependent crosstalk of GTPases with aPKC isoforms which form part of the PAR complex[28]. Consistently, Cdc42-mediated cell polarity in migrating astrocytes occurs through activation of the aPKC isoform PKCζ[28,43]. Thus, our observation of reduced sinusoidal localization of MKs in wortmannin-treated mice and mice deficient in the aPKC- isoform PKCι is supportive of a role of PI3K signalling for Cdc42/RhoA-controlled MK polarization and localization.

During the last years, experimental evidence has accumulated suggesting that MKs possess an intrinsic network involving the

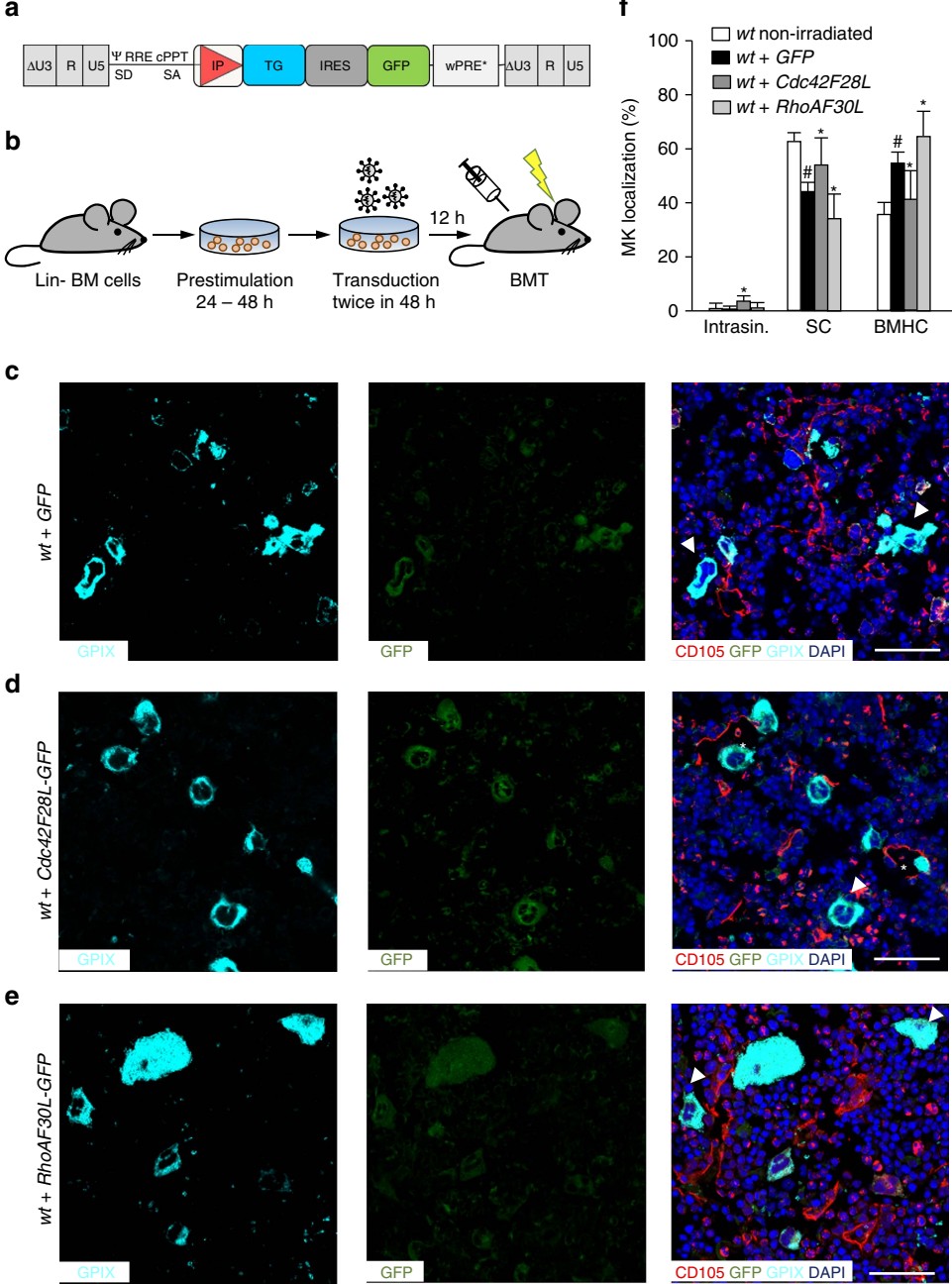

**Figure 7 | Activation states of RhoA and Cdc42 have opposing functions in MK localization.** (**a**) Third-generation self-inactivating lentiviral vectors were generated, expressing Cdc42F28L and RhoAF30L, respectively, together with GFP through an IRES, or just GFP under control of the MK-specific human *GP6* promoter in the internal position. (**b**) Work-flow. Lineage marker-negative (Lin − ) BM cells were isolated, pre-stimulated for 24 h and transduced twice. Cells were then transplanted into conditioned recipient mice. (**c–e**) Representative confocal images of immunostained BM of *wt* mice after BM transplantation of *wt* HSC transduced with GFP (**c**), constitutive active Cdc42 (F28L) (**d**) or constitutive active RhoA (F30L) (**e**). Scale bars, 50 μm. MKs, proplatelets and platelets are shown by GPIX staining in cyan colour. Endoglin staining (red) labels vessels. DAPI, blue. Arrowhead indicates MKs in BMHC, asterisk indicates intrasinusoidal MKs. (**f**) Quantification of MK localization in the BM ($n = 4$). Quantification of MK localization in non-irradiated *wt* mice (white) are shown to demonstrate altered localization after irradiation and transplantation of *wt* cells. Bar graphs represent mean ± s.d. Two-way ANOVA with Bonferroni correction for multiple comparisons. *$P < 0.05$, compared to *wt* + GFP; #$P < 0.05$, compared to *wt* not transplanted.

GPIb-IX-V complex and multiple cytoskeletal proteins including Filamin A, non-muscle myosin IIa (NMM-IIa), β-tubulin, α-actinin and Diaphanous Related Formin 1 (DIAPH1) that control the late stages of MK maturation with defects in individual proteins all giving rise to macrothrombocytopenia[40,44]. Interaction of the cytoplasmic domain of GPIbα with Filamin A provides a major link between the receptor and the MK cytoskeleton and coordinated expression of GPIbα and Filamin A is required for proper localization of either protein[45]. Deficiency of Filamin A in MKs was shown to affect DMS formation[46] and results in premature release of large, fragile platelets in humans and mice[47,48]. Together with the observation that polarity in osteoclasts depends on a network involving both Filamin A and Cdc42 (ref. 49) this points to a critical role of Filamin A for the transmission of signalling between GPIbα and Cdc42.

### a
Wild-type

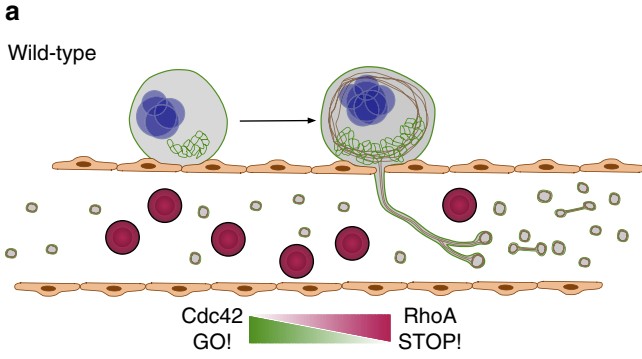

Cdc42 GO! RhoA STOP!

### b
GPIbα/Cdc42 signalling impaired

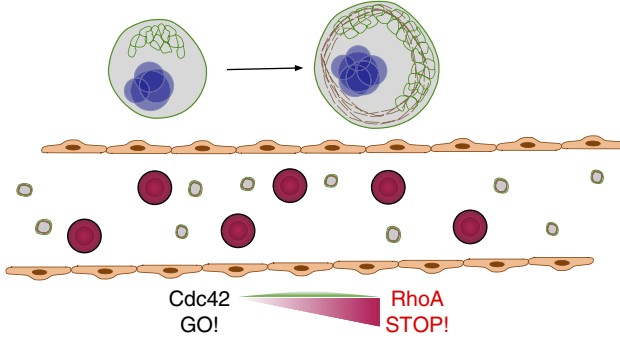

Cdc42 GO! RhoA STOP!

### c
RhoA signalling impaired

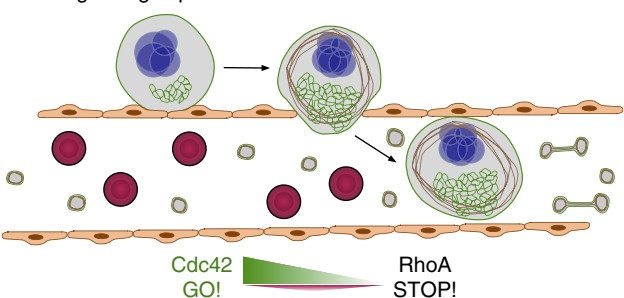

Cdc42 GO! RhoA STOP!

**Figure 8 | Model of the Cdc42/RhoA regulatory circuit controlling GPIbα-dependent platelet biogenesis.** (**a**) Wt BM MKs polarize their DMS as prerequisite for directed proplatelet release into sinusoidal vessels. This polarization is driven by locally active Cdc42. (**b**) Perturbed intracellular GPIbα-signalling reduces local Cdc42 activity and results in defective DMS polarization, reduced MK localization at BM sinusoids and decreased transendothelial proplatelet formation. (**c**) In the absence of RhoA, Cdc42 activity is locally increased resulting in hyperpolarized MKs entirely transmigrating through the endothelial barrier into sinusoidal vessels.

Our data from RhoA-deficient mice clearly exclude that size restrictions limit MK transmigration through BM sinusoids and emphasize RhoA as a critical regulator of proplatelet extension across the endothelial barrier by providing a 'stop' signal for MK polarization and thus transmigration (Fig. 8). In line with this, it has been described that excessive RhoA activation in BM-derived macrophages leads to decreased polarization, motility and cell spreading[50]. Of note, RhoA-dependent activation of Rho kinase (ROCK) and NMM-IIa was shown to negatively regulate proplatelet formation in MKs derived from CD34+ cells *in vitro*[33]. Mutations in the gene encoding NMM-IIa, *MYH9*, affect platelet biogenesis in patients with MYH9-related disorders[51]. Interestingly, MKs from mice

with MK-restricted NMM-IIa deficiency ($Myh9^{-/-}$) exhibit increased proplatelet formation *in vitro*[52] and a significant number of MK nuclei was observed in lungs of $Myh9^{-/-}$ animals compared to the wild-type[53]. However, given the multiple intrinsic defects present in $Myh9^{-/-}$ MKs, further studies using different experimental approaches will be required to decipher the role of NMM-IIa in RhoA-induced MK polarization and platelet biogenesis.

In summary, the herein described Cdc42/RhoA regulatory circuit downstream of GPIb ensures a tight control of transendothelial platelet biogenesis in the BM and opens new avenues to treat both inherited and acquired forms of thrombocytopenia that are major causes of bleeding in man, and additionally to modulate *in vitro* production of platelets in the field of transfusion medicine.

## Methods

**Mice.** All animal experiments, except studies involving BM transplantation, were carried out in the Institute of Experimental Biomedicine (Würzburg), and approved by the district government of Lower Frankonia (Bezirksregierung Unterfranken). BM transplantation experiments were carried out at the Paul-Ehrlich-Institute (Langen) and approved by the local ethics committee of the Lower Saxony State Office (according to §15 German animal protection law). $Gp1ba^{-/-}$ mice (C57Bl/6 background), $Gp1ba^{-/-;Tg}$ ($Gp1ba$-tg) mice (C57Bl/6 background), where the ectodomain of GPIbα is replaced by that of the human interleukin-4 receptor α (IL-4Rα), and $Prkci^{-/-}$ mice (C57Bl/6 background) have been described[24,54]. MK/platelet-specific conditional RhoA- and Cdc42-deficient mice (both mixed C57Bl/6 and SV129 background), carrying the Cre-recombinase under the platelet factor 4 (Pf4) promoter[55], have been described[13,14]. $RhoA^{-/-}$ mice were intercrossed with $Gp1ba^{-/-;Tg}$ mice to generate $RhoA^{-/-}/Gp1ba^{-/-;Tg}$ mice, with $Cdc42^{-/-}$ mice to generate $RhoA/Cdc42^{-/-}$ mice, and with $Rac1^{-/-}$ mice (mixed C57Bl/6 and SV129 background) to generate $RhoA/Rac1^{-/-}$ mice. $Prkci^{-/-}$ mice were kindly provided by Dr Michael Leitges[54]. $Vwf^{-/-}$ mice (C57Bl/6 background) were kindly provided by Dr Cecile Denis[56]. Mice used in experiments were 8 to 16 weeks old and sex-matched (both sexes used), if not stated otherwise.

**Antibodies and reagents.** Anaesthetic drugs (medetomidine (Pfizer), midazolam (Roche) and fentanyl (Janssen-Cilag)) were used according to the regulations of the district government of Lower Frankonia (Bezirksregierung Unterfranken). Mice were bled into high-molecular-weight heparin (Ratiopharm). Mice were treated with the selective PI3K inhibitor wortmannin (Sigma-Aldrich; $1 \mu g g^{-1}$ body weight; days 1 and 3), the microtubule polymerization inhibitor nocodazole (Sigma-Aldrich; $10 \mu g g^{-1}$ body weight; days 1–3), or with monovalent Fab fragments of antibodies directed against the major ligand-binding domain of GPIbα (p0p/B-Fab (ref. 25); 100 μg per mouse, days 1 and 3), integrin αIIbβ3 (JON/A (ref. 36); 100 μg per mouse, days 1–3) or GPV (DOM1 (ref. 37); 100 μg per mouse, days 1–3). Mice were killed and femora dissected and processed on day 5. Antibodies were fluorescently labelled using protein labelling kits (Alexa Fluor 488/546/647, Life Technologies). Anti-CD105 antibody (120402 (MJ7/18), BioLegend) or Tetramethylrhodamine dextran (2 MDa; Thermo Scientific) were used for BM vasculature staining. All other antibodies were produced and modified in our laboratory[37]. Antibodies were incubated for 6–8 h with immobilized papain or for 24 h with immobilized pepsin according to the manufacturer's instructions (Pierce Biotechnology, Inc.), and the preparations were then applied to an immobilized protein A column, followed by an immobilized protein G column (Pharmacia) to remove Fc fragments and undigested IgG. Purity of Fab or F(ab)2 fragments was tested by SDS–PAGE. For control experiments, purified rat IgG2a (Serotec) and nonimmune control rat IgG Fab were used.

**Immunofluorescence staining on whole femora cryosections.** Femora were isolated, fixed with 4% PFA (AppliChem) and 5 mM sucrose (Sigma-Aldrich), transferred into 10% sucrose in PBS and dehydrated using a graded sucrose series. Subsequently, the samples were embedded in Cryo-Gel (Leica Biosystems) and shock frozen in liquid nitrogen. Frozen samples were stored at −80 °C. Seven-micrometer-thick cryosections were generated using the CryoJane tape transfer system (Leica Biosystems) and probed with Alexa488-conjugated anti-GPIX antibody, to specifically label platelets and MKs, and Alexa647-conjugated anti-CD105 antibodies (3.33 mg ml⁻¹, 120402 (MJ7/18), Biolegend) to stain the endothelium. Nuclei were stained using DAPI (4,6-diamidino-2-phenylindole; 1 mg ml⁻¹, D1306, Invitrogen). Samples were visualized with a Leica TCS SP5 confocal microscope (Leica Microsystems).

**TEM of bone marrow megakaryocytes.** For TEM of MKs, BM was flushed from 12- to 16-week-old mice using Karnovsky fixative (2% PFA, 2.5% glutaraldehyde

in 0.1 M cacodylate buffer) and incubated overnight at 4 °C. Subsequently, fatty components of the samples were fixed with 2% osmium tetroxide in 50 mM sodium cacodylate (pH 7.2), stained with 0.5% aqueous uranyl acetate, dehydrated with a graded ethanol series and embedded in Epon 812. Ultra-thin sections were stained with 2% uranyl acetate (in 100% ethanol) followed by lead citrate. Images were taken on a Zeiss EM900 TEM (Zeiss). ImageJ software (Wayne Rasband, NIH, USA: http://rsb.info.nih.gov/ij) was used to quantify DMS.

**Two-photon intravital microscopy of the BM.** Mice were anaesthetized by intraperitoneal injection of medetomidine 0.5 µg g$^{-1}$, midazolam 5 µg g$^{-1}$ and fentanyl 0.05 µg g$^{-1}$ body weight. A 1 cm incision was made along the midline to expose the frontoparietal skull, while carefully avoiding damage to the bone tissue. The mouse was placed on a customized metal stage equipped with a stereotactic holder to immobilize its head. BM vasculature was visualized by injection of tetramethylrhodamine dextran (8 µg g$^{-1}$ body weight, 2 MDa, Thermo Scientific). Platelets and MKs were antibody stained with anti-GPIX-AlexaFluor 488 (i.v. injection of 0.6 µg g$^{-1}$ body weight). Images were acquired with a fluorescence microscope equipped with a × 20 water objective with a numerical aperture of 0.95 and a TriM Scope II multiphoton system (LaVision BioTec), controlled by ImSpector Pro-V380 software (LaVision BioTec). Emission was detected with HQ535/50-nm and ET605/70-nm filters. A tunable broad-band Ti:Sa laser (Chameleon, Coherent) was used at 760 nm to capture Alexa Fluor 488 and rhodamine dextran fluorescence. ImageJ software (NIH) was used to generate movies.

**Flow cytometry.** Diluted blood (1:20) was incubated for 15 min at room temperature (RT) with fluorophore-conjugated antibodies (2 µg ml$^{-1}$) directed against platelet surface GPs[57]. Platelet count and size (forward scatter) were assessed using a FACSCalibur (BD Biosciences) flow cytometer.

**Immunoblotting.** Denatured platelet lysates were separated by SDS–PAGE and blotted onto polyvinylidene difluoride membranes. Membranes were incubated with anti-RhoA antibody (0.5 µg ml$^{-1}$, Cytoskeleton Inc.), anti-Cdc42 antibody (0.5 µg ml$^{-1}$, Cytoskeleton Inc.) or anti-Rac1 antibodies (0.5 µg ml$^{-1}$, BD Biosciences) antibodies followed by incubation with appropriate horseradish peroxidase-conjugated secondary antibodies (1 h, room temperature) and enhanced chemiluminescence solution (JM-K820-500, MoBiTec). Images were recorded using a MultiImage II FC Light Cabinet (Alpha Innotech Corporation) device. As loading control, integrin β3 (GPIIIa) levels were determined.

**Histology.** Three-micrometer-thick sections of formalin-fixed paraffin-embedded spleens and lungs were prepared, deparaffinized and stained with haematoxylin and eosin (MHS32 and 318906, Sigma-Aldrich). MKs were stained with HRP-labelled anti-GPIbα antibodies after antigen retrieval and detected with 3-amino-9-ethylcarbazole substrate. MK number, morphology and localization were analyzed with an inverted Leica DMI 4,000 B microscope.

**In vitro BM MK isolation and differentiation.** BM cells were obtained from femur and tibia by flushing, and lineage depletion (Lin −) was performed using CD16/CD32 +, Gr1 +, B220 + and CD11b + antibodies (Biolegend). Lin − cells were cultured in 2.6% nutrient-supplemented StemPro medium with 2 mM L-glutamine, 100 IU ml$^{-1}$ penicillin, 50 mg ml$^{-1}$ streptomycin and 20 ng ml$^{-1}$ murine stem cell factor, 15 IU ml$^{-1}$ heparin and 50 ng ml$^{-1}$ of mTPO at 37 °C under 5% CO$_2$ for 3 days. Where indicated, MK cultures were supplemented daily with 50 µg ml$^{-1}$ of p0p/B-Fab fragment. Mature MKs were enriched by a one-step BSA gradient.

**Analysis of MK polarization.** For immunofluorescence, MKs in suspension were fixed and permeabilized in one step for 30 min in PBS with 3.7% formaldehyde and 0.05% Triton X-100. Samples were saturated in 1% fatty acid free BSA in PBS. Incubation with antibodies, fluorescent secondary antibodies, AlexaFluor-labelled phalloidin or DAPI was performed for 1 h at RT. For 3D imaging, cells were kept in µ-slide ibiTreat chambers (Ibidi) in PBS. Confocal images were captured with a LSM780 operated with Zen software using a × 63, 1.4 NA Plan Apochromatic objective lens (Carl Zeiss). Profiling of fluorescence intensity was done with ImageJ (NIH). For 3D-analysis, z-stacks were taken and processed with the Imaris 6.4.2 software (Bitplane AG).

**FRET-based measurement of Cdc42 activity.** The Raichu-Cdc42 encoding sequence was amplified by PCR using Raichu-1054X plasmid as a template and cloned in frame into the pTRIP-IRES lentiviral vector plasmid using BamHI and NheI restriction sites (underlined)[32]. The primers were 5′-CGCGGATCCTTGG CAAGAATTCGGCATGG-3 (forward) and 5′-CTAGCTAGCGGCAGAGGGAAA AAGATCCGTCGAC-3′ (reverse). UT711oc1 cells were transduced by incubation with lentiviral particles at a multiplicity of infection (MOI) of 1 for 2 days. Transduction efficiency was checked by fluorescence and reached about 100% of the population. Transduction of primary MKs was performed on Lin − population

at an MOI of 5 for 1 day, then 50 ng ml$^{-1}$ mTPO were added for 3 additional days. FRET efficiency was represented as the colour-coded ratio image of YFP/CFP after background subtraction using MetaMorph (Universal Imaging). Acquisitions were performed with a LSM780 confocal microscope piloted by the Zen software, using a − 63, 1.4 NA Plan Apochromatic lens (Carl Zeiss).

**Lentiviral gene transfer of Rho GTPase mutants.** The cDNAs of Cdc42F28L or RhoAF30L were inserted into lentiviral vectors upstream of an IRES co-expressing eGFP as reporter (lentiviral vector RRL.PPT.SF.eGFP.pre kindly provided by A. Schambach, Hannover Medical School[58]). Expression was controlled by the MK-specific human GP6 promoter (hGP6, − 697 to + 29 (ref. 59)). Lentiviral particles were produced by transient transfection in 293T cells. The day before transfection, 5 × 10$^6$ 293T cells were seeded in 10 cm plates in DMEM medium containing 10% fetal calf serum, 1% glutamine, 1% penicillin/streptomycin, 20 mM HEPES. The next day, 293T cells were transfected by calcium-phosphate transfection method with 10 µg of vector plasmid, 10 µg gag/pol plasmid (pCDA3.GP.4XC plasmid), 1.5 µg plasmid encoding the VSV-G envelope (pMD.G plasmid), 5 µg Rev encoding plasmid in DMEM medium supplemented with 25 µM of chloroquine. Lentiviral supernatants were collected after 36 and 48 h post transfection, and concentrated by ultra-speed centrifugation at 80,000 g × 2 h. Concentrated viral particles were resuspended in StemSpan medium (Stem Cell Technologies, Köln, Germany) and stored at − 80 °C. Prior to viral transduction, Lin − BM cells were pre-stimulated for 24–48 h in StemSpan containing 10 ng ml$^{-1}$ murine stem cell factor, 20 ng ml$^{-1}$ murine TPO, 10 ng ml$^{-1}$ recombinant human FGF-1, 20 ng ml$^{-1}$ murine IGF2, 1% penicillin/streptomycin, 2 mM glutamine. 5 × 10$^5$ Lin − cells were transduced twice on two following days with lentiviral vectors on Retronectin (Takara-Clontech, 10 µg cm$^{-2}$) coated wells (24-well or 12-well plates, dependent on the number of cells for transduction, with a defined MOI).

**BM transplantation.** 8–12-week-old female C57Bl/6 mice were preconditioned with 10 Gy irradiation and transplanted with 5 × 10$^5$ cells. Cell infusion was performed via tail vein injection in a final volume of 150 µl. All mice were kept in the specified pathogen-free animal facilities of the Paul-Ehrlich-Institute, Langen, Germany.

**Statistical analysis.** When comparing two experimental groups, data distribution was analysed using the Shapiro–Wilk test. Where indicated, statistical significance between two experimental groups was analysed using an unpaired two-tailed Student's t-test. Otherwise, data were analysed using two-way analysis of variance (ANOVA) with Bonferroni correction for multiple comparisons (Prism 7; GraphPad Software). P-values < 0.05 were considered as statistically significant. *P < 0.05; **P < 0.01; ***P < 0.001 or as otherwise stated. Data are presented as mean ± s.d.

**Data availability.** All data generated or analysed during this study are included in this published article (and its Supplementary Information files).

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

## Acknowledgements

We thank Jonas Müller, Stefanie Hartmann, Sylvia Hengst and Franziska Schenk for excellent technical assistance and Imke Meyer for initial support during multi-photon microscopy. We are also thankful to the microscopy platform of the Bioimaging Centre (Rudolf Virchow Centre) and of the INSERM UMR1048, especially E. Vega, for providing technical infrastructure and support. This work was supported by the Deutsche Forschungsgemeinschaft (NI 556/9-1 to B.N., SFB688 to D.S. and K.G.H., and PL 707/2-1 to I.P.) and the Rudolf Virchow Center. S.S., J.M.M.v.E., D.C. and A.A.B. were supported by a grant of the German Excellence Initiative to the Graduate School of Life Sciences, University of Würzburg. Development of mice in Z.M.R. laboratory was supported by NIH grant HL-117722 and MERU Foundation, Italy.

## Author contributions

S.D. designed research, performed experiments, analysed data and wrote the manuscript. M.P., A.A., T.H., K.A., F.G. and N.H. performed experiments and analysed the data. D.S., S.S., D.C., P.N., U.M., I.P., and F.G.-I. performed experiments, analysed data and wrote the manuscript. J.M.M.v.E. and A.A.B. performed experiments. O.A., M.G. and K.G.H. analysed the data. G.K., A.T.N. contributed to the writing of the manuscript. H.S., J.W., C.B. and Z.M.R. provided vital new reagents and contributed to the writing of the manuscript. B.N. conceived the study, designed research, analysed data and wrote the manuscript.

## Additional information

**Competing interests:** The authors declare no competing financial interests.

