## [Peer Review File · Nature Communications]

Reviewers' comments:

Reviewer #1 (Remarks to the Author):

General Comments to Authors

In this interesting manuscript the authors provide a detailed account of the distribution of megakaryocytes in the mouse bone marrow and delineate a mechanism whereby GPIb-alpha and two small GTPases, Cdc42 and RhoA, control the positioning of megakaryocytes in the bone marrow and the release of proplatelets. The bulk of the experimental data reflect analysis of the position of megakaryocytes, i.e. the percentage present in the sinusoid lumen, or in close juxtaposition to the sinusoid wall, or more distant from the sinusoids into the bone marrow hematopoietic compartment. This is done in WT mice without/with specific pharmacologic treatments and in a number of mutant mouse strains, including the one with deletion of the GPIb-alpha ectodomain and the Cdc42 and RhoA KO's. On the basis of obtained results, the authors conclude that GPIb signaling in concert with Cdc42 function is required for the correct positioning of megakaryocytes in close proximity of sinusoids and for proplatelet release, processes negatively regulated by RhoA.

The observations presented in the paper are solid and interesting; unfortunately, there is no attempt to provide a mechanistic explanation for the proposed crucial functions assigned to the extracytoplasmic GPIb-alpha domain and the intracytoplasmic molecular checkpoint operated by Cdc42/RhoA. There is no mention in the paper of GPIb ligands beyond VWF and no characterization of the antibody that produces a phenotype mimicking lack of the GPIb-alpha ectodomain. These deficiencies leave the sense of a potentially outstanding but incomplete work. The Discussion, in particular, remains predominantly speculative

Specific Comments

There is literature on the role of GPIb cytoplasmic domain and its interaction in determining platelet production and size. Not clear why this has been ignored in the discussion or as hypothesis to test.

Likewise, much is predicated on the observations made with the anti-GPIb-alpha antibody. Not clear why its effects on known GPIb functions have not been characterized in more depth.

The authors should have looked at other mouse strains that express human GPIb-alpha transgenically as more appropriate controls for their work than WT BL6 mice.

The authors fail to mention whether the mechanism of MK migration regulated by GPIb-Cdc42/RhoA is associated with the SDF-1/CXCR4 axis or not. Cdc42 is required for exocytosis and some of these authors previously showed enhanced alpha-granule secretion and P-selectin expression in Cdc42^{-/-} platelets. SDF-1 has been known to play an important role in MK migration and alpha-granule is one of the source that stores this chemokine. It would be relevant to discuss if genetic ablation of Cdc42 and/or RhoA affects SDF-1 secretion or CXCR4 expression.

The description of statistical analysis is incomplete and superficial. In the case of ANOVA, the post test used for group comparisons is not mentioned. Whether the use of ANOVA and student's t test was justified based on distribution and variance was not addressed. If indeed the error bars show 1 SD, some of the p values shown look frankly difficult to rationalize.

Reviewer #2 (Remarks to the Author):

The authors have extensively studied the role of the small GTPases Rac1, Cdc42 and RhoA in platelet formation and function (Pleines et al. Blood 2012; Blood 2013). Recent studies have also focused on the role of Cdc42 in MK maturation (Antkowiak et al. JTH 2016; Palazzo et al. JTH 2016). Thus, the novelty of the manuscript resides primarily in the fact that GPIIb₃ signals to Cdc42 and that RhoA affects GPIIb₃-Cdc42 signaling. The study could benefit from additional controls such as immunotargeting other platelet/MK glycoproteins and using other macrothrombocytopenic mouse models.

Criticisms:

- 1) It is distracting to start a paper with a supplemental figure. The authors should reconsider to add some of the results presented in the supplementary figures into main figures.
- 2) The use of wortmannin in mice can have an array of side targets, not just megakaryocytes. For example, did the authors verify that sinusoidal cells are unaffected by wortmannin?
- 3) In Figure 1h, p0p/B-Fab induces a ~25% decrease in blood platelet counts. The authors have shown previously that p0p/B-Fab does not affect platelet counts (Kleinschnitz et al. Circulation 2007). How do they explain the discrepancy? Further, a decrease in blood platelet counts due to p0p/B-Fab-induced platelet clearance is expected to indirectly promote platelet production and therefore affect BM MK morphology and localization. What is the primary target of p0p/B-Fab: platelet or MK GPIIb₃? Controls such as Fab fragments of antibodies directed against αIIbβ₃ that lead to reduction in platelet counts should be used.
- 4) The authors wish to understand the thrombocytopenia of patients with Bernard-Soulier syndrome, a bleeding disorder caused by the absence of GPIIb-IX. Thus, the rationale for studying Gp1b-Tg mice rather than Gp1b^{-/-} mice is unclear. Is Cdc42 GTP loading decreased in Gp1b^{-/-} MKs?
- 5) The authors should speculate more on which extracellular cues GPIIb₃ uses to activate Cdc42 and RhoA.
- 6) Several other mouse models develop macrothrombocytopenia such as Myh9^{-/-} and Flna^{-/-} mice. Are these also related to decreased GPIIb₃-Cdc42 signaling?
- 7) The finding that RhoA^{-/-} MKs transverse the endothelium easily suggests that integrin activity should be almost abolished- or not important at this step. This should be at least discussed.

Reviewer #3 (Remarks to the Author):

This study describes the role of the small GTPases Rho A and Cdc42 and the von Willebrand factor receptor in the generation of platelets. The manuscript is based on very nice previous mouse knockout studies that clearly implicate small GTPases in production of thrombocytes. The authors take advantage of knockout mouse models, live animal imaging, and inhibition of signaling pathways with drugs/antibodies to test the idea that the vWf receptor links to small GTPases to control megakaryocyte polarization during the generation of thrombocytes. This paper has some beautiful imaging and high quality data, but there are also major concerns with their model and some of the experiments.

1. One of the major concerns with the author's central conclusion is that BOTH humans that do not have von Willebrand factor and mice lacking VWF appear to have normal platelet counts. And I believe the data clearly shows that the distribution of the megakaryocytes is not disturbed in these models. This creates a major hole in their model.
2. Another major concern with this study is the lack of alternative hypothesis and controls in experiments. Treating mice with wortmannin for 5 days is likely to have a very broad effect on the animal and create many other side effects. Again, treating mice with an anti-platelet antibody will

cause an immune thrombocytopenia and a cytokine response, leading to a very complex phenotype.

3. The authors need to do a better job showing that this biological event involves a connection between the vWf and the GTPases.

4. It appears that most of the studies have only been done in mouse models. The manuscript lacks human data that calls into question the medical relevance of the studies.

Response letter to the reviewers

Reviewer 1:

1. There is literature on the role of GPIb cytoplasmic domain and its interaction in determining platelet production and size. Not clear why this has been ignored in the discussion or as hypothesis to test.

We agree with the reviewer that we should have discussed the role particularly of the GPIb/Filamin A interaction for platelet production and sizing in more detail. We have now included a paragraph discussing the importance of this interaction for platelet production (Discussion, p. 13) and also highlighted the possible relevance for signal transmission from GPIb to Cdc42.

2. Likewise, much is predicated on the observations made with the anti-GPIb-alpha antibody. Not clear why its effects on known GPIb functions have not been characterized in more depth.

We apologize for not having described and discussed the herein used p0p/B Fab in more detail. Indeed, it has been shown that the p0p/B Fab blocks the thrombin binding site in GPIbalpha (Ruggeri and co-workers). This information is now included in the new Supplementary Fig. 10c and discussed on p. 11 of the revised manuscript (see query 3 below).

Furthermore, in line with our findings (Figure 2d,e) it has been shown by Kleinschnitz *et al* (Circulation 2007) that *in vivo* injection of p0p/B Fab does not lead to acute thrombocytopenia and the Fab has been used in numerous studies to block GPIb α in acute disease models without affecting peripheral platelet counts. In all these experiments, the observation period was less than 24 hours (e.g. Schuhmann *et al.*, J Neuroinflammation 2017; Pachel *et al.*, ATV 2016).

In contrast, *in vivo* treatment with the p0p/B-IgG (and other GPIb antibodies or their F(ab)₂ fragments) was shown to result in an acute thrombocytopenic phase and to affect MK morphology during the recovery phase, demonstrating its direct effect on MKs (Poujol *et al.*, Blood 2003). This strongly suggests that the here observed slight effect on platelet counts upon repeated injection of p0p/B (Figure 2d,e) is indeed caused by Fab binding to MK GPIb, leading to impaired signaling of the receptor.

Together with the observation that repeated *in vitro* treatment of MKs with p0p/B Fab has a pronounced effect and reverses the hyperpolarization of RhoA KO MKs (new Figure 5g), our results support the hypothesis that GPIb α -mediated regulation of MK polarization and migration might be a cell intrinsic process which can be modulated by altering GPIb membrane localization/clustering and thereby, possibly, its signaling by p0p/B-Fab treatment. These considerations have been added to the discussion (Discussion, p. 12).

3. The authors should have looked at other mouse strains that express human GPIb-alpha transgenically as more appropriate controls for their work than WT BL6 mice.

As requested, we have analyzed MK localization in the BM in mice expressing either human wt GPIb α (hGPIb α) or a mutant version in which the thrombin site is mutated (hGPIb α D277N). We found MK localization was similar in both genotypes, indicating that thrombin binding is not required for GPIb-mediated signaling in MK polarization and localization. These results are presented in the new Supplemental Figure 10d,e and discussed on p. 11.

4. The authors fail to mention whether the mechanism of MK migration regulated by GPIIb-Cdc42/RhoA is associated with the SDF-1/CXCR4 axis or not. Cdc42 is required for exocytosis and some of these authors previously showed enhanced alpha-granule secretion and P-selectin expression in Cdc42^{-/-} platelets. SDF-1 has been known to play an important role in MK migration and α -granule is one of the source that stores this chemokine. It would be relevant to discuss if genetic ablation of Cdc42 and/or RhoA affects SDF-1 secretion or CXCR4 expression.

We admit that the effect of SDF-1/CXCR4 signaling on MK polarization/migration was not a focus of our work. The reason for this is that there is another study from our group on the role of MK migration in platelet biogenesis, in which the reviewer's question was addressed. This work is currently in revision and we hope that the reviewer understands that this confidential information cannot be included as a result in our paper.

[UNPUBLISHED DATA REDACTED BY EDITORIAL TEAM AS PER AUTHORIAL REQUEST]

5. The description of statistical analysis is incomplete and superficial. In the case of ANOVA, the post test used for group comparisons is not mentioned. Whether the use of ANOVA and student's t test was justified based on distribution and variance was not addressed. If indeed the error bars show 1 SD, some of the p values shown look frankly difficult to rationalize.

We thank the reviewer for pointing out this issue. We have revised the statistics as follows (see Methods):

When comparing two experimental groups, data distribution was first analyzed using the Shapiro-Wilk test. Statistical significance between two experimental groups was then analyzed using an unpaired two-tailed Student's *t* test or a Mann Whitney test. When more than two experimental groups were compared, data were analyzed using 2-way analysis of variance (ANOVA) with Bonferroni correction for multiple comparisons (Prism 7; GraphPad Software). P-values <0.05 were considered as statistically significant. **P*<0.05; ***P*<0.01; ****P*<0.001 or as otherwise stated. Data are presented as mean \pm S.D.

This information is included on p. 18 (Methods section). The notion that the p-values in the shown graphs are often very low is true and due to the fact that in 2-way Anova means but not the distribution of data are compared.

Reviewer #2 (Remarks to the Author):

Criticisms:

1) It is distracting to start a paper with a supplemental figure. The authors should reconsider to add some of the results presented in the supplementary figures into main figures.

We re-arranged the figures and now included the analysis of the BM MK localization in GPIIb α -deficient mice as new Figure 1 (see below).

2) The use of wortmannin in mice can have an array of side targets, not just megakaryocytes. For example, did the authors verify that sinusoidal cells are unaffected by wortmannin?

We agree with the reviewer that the use of wortmannin as a PI3K inhibitor possibly resulted in side effects on cells other than MKs. We have therefore moved these results to the Supplement as new Supplemental Figure 2, included TEM pictures to visualize the MK morphology after treatment, and furthermore toned down the conclusions drawn from this experiment (Discussion p. 12). However, we do still believe that it is valuable mentioning the wortmannin results, particularly with respect to the altered localization of BM MKs in mice lacking PKC ζ , a known downstream target of PI3K signaling (revised Supplementary Figure 3a,b)

3) In Figure 1h, p0p/B-Fab induces a ~25% decrease in blood platelet counts. The authors have shown previously that p0p/B-Fab does not affect platelet counts (Kleinschnitz *et al.* *Circulation* 2007). How do they explain the discrepancy? Further, a decrease in blood platelet counts due to p0p/B-Fab-induced platelet clearance is expected to indirectly promote platelet production and therefore affect BM MK morphology and localization. What is the primary target of p0p/B-Fab: platelet or MK GPIIb α ? Controls such as Fab fragments of antibodies directed against α IIb β 3 that lead to reduction in platelet counts should be used.

The reviewer makes an important point. However, in the Kleinschnitz paper, platelet counts were only determined and found to be normal 1 h after p0p/B Fab injection. This is in line with our results of unaltered platelet counts on day 1 of p0p/B Fab treatment (revised Figure 2d,e; left bar graph). We only observed decreased platelet counts after repeated injection of p0p/B Fab (day 5, Figure 2d,e; right bar graph).

As described above (Reviewer 1, query 2), it is important to note that the p0p/B Fab does not induce platelet clearance and acute thrombocytopenia upon *in vivo* injection, but just blocks GPIIb on circulating platelets. This has been confirmed in many independent studies. The pronounced antithrombotic/anti-inflammatory effect of this platelet GPIIb α blockade has been described by Kleinschnitz *et al.* (*Circulation* 2007). In contrast, anti-GPIIb-IgG or F(ab)₂ fragments thereof cause pronounced thrombocytopenia, presumably by dimerizing GPIIb on the platelet surface. In addition, the effects of p0p/B IgG binding to MKs were described by Poujol *et al.* (*Blood* 2003). As stated above, the effect of p0p/B Fab *in vivo* treatment on MK may be explained by alteration of GPIIb localization/movement in the membrane which may affect its signaling and/or clustering.

Lastly, as requested by the reviewer, we have analyzed the effect *in vivo* treatment of WT and RhoA KO mice with the JON/A Fab and DOM1 Fab leading to blockade of integrin $\alpha\text{IIb}\beta\text{3}$ and occupancy of GPV, respectively (Nieswandt *et al.* J Exp Med 2001 and Blood 2000). Notably, these antibody Fab fragments did not cause altered MK localization in the WT, nor were they able to revert the transendothelial migration of RhoA KO MKs emphasizing that treatment with p0p/B Fab affects a specific GPIb-dependent signaling response in MKs. These results are presented in the new Supplementary Fig. 7b,c.

4) The authors wish to understand the thrombocytopenia of patients with Bernard-Soulier syndrome, a bleeding disorder caused by the absence of GPIb-IX. Thus, the rationale for studying Gp1b-Tg mice rather than Gp1b^{-/-} mice is unclear. Is Cdc42 GTP loading decreased in Gp1b^{-/-} MKs?

The reviewer has a point. The reason we initially focused on the Gp1ba-Tg mice for our studies is that MKs of these mice do not exhibit the complex defects present in the KO MKs. As requested, we have analyzed the MK distribution in the BM of Gp1ba KO mice and found a reduction in the proportion of MKs with sinusoidal contact (SC), which was comparable to that seen in Gp1ba-Tg mice. These results are shown in Figure 1 of the revised manuscript:

Since we had to newly generate the Gp1ba KO mouse line by backcrossing from the *GP1ba-Tg* line for this revision we have not yet obtained sufficient animals to be able to investigate whether Cdc42 GTP loading is decreased in Gp1ba KO MKs.

5) The authors should speculate more on which extracellular cues GPIba uses to activate Cdc42 and RhoA.

We agree that we should have discussed this issue in more detail initially. We have now expanded this part of the discussion. Our results of normal MK distribution in vWF KO mice and hGPIb-D277N mice (see Reviewer 1, query 3) do not indicate that vWF or thrombin play a major role in GPIb-mediated regulation of MK polarization and migration. In fact, our new results on the effect of p0p/Fab on hyperpolarization on RhoA KO MKs *in vitro* (new Figure

5g) strongly suggests that GPIIb-mediated MK polarization/migration is a cell intrinsic process which may not require binding of an extracellular ligand (see response to query 2, Reviewer 1).

6) Several other mouse models develop macrothrombocytopenia such as *Myh9*^{-/-} and *Flna*^{-/-} mice. Are these also related to decreased GPIIb α -Cdc42 signaling?

The reviewer makes an important point. *FlnA* and *Myh9* (NMM-IIa) are important downstream signaling proteins of GPIIb and RhoA in MKs, respectively, and we plan detailed studies on the involvement of these proteins in GPIIb-Cdc42/RhoA controlled MK polarization and migration in the future. For now we were able to analyze MK localization in the BM of MK-specific *Myh9* KO mice (see figure below). We found an increased proportion of clustered MKs in direct contact with the BM sinusoids (SC). In addition, as described in Pertuy *et al.*, 2014 we could observe MK fragmentation into the sinusoids and a slightly increased number of intrasinusoidal MKs compared to the control. However, since *Myh9* KO MKs exhibit severe defects leading to loss of cellular integrity and fragmentation (Eckly *et al.* Blood 2009) it is difficult to draw direct conclusions on the role of GPIIb-Cdc42 or RhoA in this process without raising a number of new questions. We have therefore decided to confidentially show these results here but to not include them into the revised version of the manuscript.

7) The finding that *RhoA*^{-/-} MKs transverse the endothelium easily suggests that integrin activity should be almost abolished- or not important at this step. This should be at least discussed.

We agree that it will be of high interest to investigate a potential role of integrins in the process of GPIIb-controlled MK polarization and platelet biogenesis and this issue will be part of our planned future studies. While we can only speculate at present, our finding that *in vivo* treatment of *RhoA* KO mice with Fab fragments blocking the major integrin α IIb β 3 does not alter numbers of intrasinusoidal MKs indicates that at least the α IIb β 3 integrin does not play an important role during this process. However, clearly further detailed studies are required to address a potential crosstalk of GPIIb α and the different integrins expressed in MKs.

Reviewer #3 (Remarks to the Author):

This study describes the role of the small GTPases Rho A and Cdc42 and the von Willebrand factor receptor in the generation of platelets. The manuscript is based on very nice previous mouse knockout studies that clearly implicate small GTPases in production of thrombocytes. The authors take advantage of knockout mouse models, live animal imaging, and inhibition of signaling pathways with drugs/antibodies to test the idea that the vWf receptor links to small GTPases to control megakaryocyte polarization during the generation

of thrombocytes. This paper has some beautiful imaging and high quality data, but there are also major concerns with their model and some of the experiments.

1. One of the major concerns with the author's central conclusion is that BOTH humans that do not have von Willebrand factor and mice lacking VWF appear to have normal platelet counts. And I believe the data clearly shows that the distribution of the megakaryocytes is not disturbed in these models. This creates a major hole in their model.

The reviewer makes an important point. It is indeed exactly the observation of normal platelet counts and normal MK distribution in the BM upon vWF deficiency leading to the interesting conclusion that vWF is most likely not a (physiological) ligand of GPIIb/IIIa required for the herein described process. Indeed, our data support the hypothesis that GPIIb-mediated MK polarization and platelet biogenesis does not require binding of an ectopic ligand but may occur via a cell-intrinsic process (see Reviewer 1, query 2 and Reviewer 2, query 3).

2. Another major concern with this study is the lack of alternative hypothesis and controls in experiments. Treating mice with wortmannin for 5 days is likely to have a very broad effect on the animal and create many other side effects. Again, treating mice with an anti-platelet antibody will cause an immune thrombocytopenia and a cytokine response, leading to a very complex phenotype.

As discussed above (Reviewer 2, query 2), we agree with the Reviewer that the use of wortmannin as a PI3K inhibitor possibly resulted in side effects on cells other than MKs. We have therefore moved these results to the Supplement and furthermore toned down the conclusions drawn from this experiment.

As described above, the Fab fragments used in this study have been repeatedly used *in vivo* and were shown not to induce acute thrombocytopenia and an inflammatory response (see Reviewer 1, query 2 and Reviewer 2, query 3).

3. The authors need to do a better job showing that this biological event involves a connection between the vWf and the GTPases.

Please see the answer to the first query.

4. It appears that most of the studies have only been done in mouse models. The manuscript lacks human data that calls into question the medical relevance of the studies.

The results presented in this manuscript were indeed obtained from mice. However, mouse models of platelet disorders, such as BSS, Glanzmann thrombasthenia, Myh9-related disorders and many others were shown to well mimic the phenotype in humans and have proven highly useful to study the underlying mechanisms leading to defective platelet biogenesis or function. On the other hand, the platelet biogenesis defect in BSS patients is extremely well characterized and was found to be perfectly reproduced in GPIIb/IIIa and GPIX mutant mice. Thus, we are confident that the mechanism of GPIIb-Cdc42-RhoA-controlled process of MK polarization and transendothelial platelet biogenesis is also operating in humans.

REVIEWERS' COMMENTS:

Reviewer #2 (Remarks to the Author):

This revised paper by Dütting et al is greatly improved and the authors answered most of the questions sufficiently.

No data have been provided concerning the role of Filamin A, the major down stream effector of GPIbalpha.

Confidential data shown in the rebuttal concerning the role of CXCL12 in the migration refer to mature megakaryocytes (MKs). It is unlikely that mature MKs migrate. Megakaryocyte progenitor data would be more to the point.

Discussion, line 316: The sentence "negatively regulate proplatelet formation in CD34+ cells " should be reformulated to negatively regulate proplatelet formation in megakaryocytes derived from CD34+ cells.

Reviewer #3 (Remarks to the Author):

The authors have adequately addressed all of my concerns.

Response letter to the reviewers_2

We would like to thank both reviewers for the quick and fair handling of our manuscript and their favorable responses.

Reviewer #2 (Remarks to the Author):

This revised paper by Dütting *et al* is greatly improved and the authors answered most of the questions sufficiently.

No data have been provided concerning the role of Filamin A, the major down stream effector of GPIbalpha.

The reviewer's point is well taken. Indeed, we had aimed to address the reviewer's comment by analyzing MK localization in the bone marrow from Filamin A-deficient mice. While we were able to obtain bone samples from a small number of mice in collaboration with Dr. Herve Falet (Blood Center of Wisconsin), the quality of the samples was not sufficiently good to enable us to draw a clear conclusion about the outcome of Filamin A deficiency on MK localization in the bone marrow. These data were therefore not included in the revised version of the manuscript. However, studying the role of Filamin A in GPIb-Cdc42/RhoA-dependent platelet biogenesis will for sure remain a focus of our future work.

Confidential data shown in the rebuttal concerning the role of CXCL12 in the migration refer to mature megakaryocytes (MKs). It is unlikely that mature MKs migrate. Megakaryocyte progenitor data would be more to the point.

The reviewer makes a point that was also raised by the reviewers of the manuscript containing the confidential data and has been addressed during the revision of that manuscript. We do not deny that the data only to a limited extent allows conclusions about the migration behavior of MK progenitors, which are, unfortunately, very difficult to visualize due to the lack of specific markers.

What we aimed to point out by showing those data in the context of this study is the observation that interfering with SDF-1/CXCR4 signaling does not affect MK numbers or their localization in the bone marrow. Thus, independent of the potential impact on migration of MK progenitors, this finding clearly counts against an involvement of the SDF-1/CXCR4 pathway in GPIb-dependent MK localization/polarization.

Discussion, line 316: The sentence "negatively regulate proplatelet formation in CD34+ cells" should be reformulated to negatively regulate proplatelet formation in megakaryocytes derived from CD34+ cells.

We have re-worded the sentence as requested.

Reviewer #3 (Remarks to the Author):

The authors have adequately addressed all of my concerns.

We are grateful to the reviewer for his positive feedback.